# The Role of Gut Microbiota in Gastrointestinal Symptoms of Children with ASD

**DOI:** 10.3390/medicina55080408

**Published:** 2019-07-26

**Authors:** Agustín Ernesto Martínez-González, Pedro Andreo-Martínez

**Affiliations:** 1Department of Developmental Psychology and Didactics, University of Alicante, 03080 Alicante, Spain; 2Department of Agricultural Chemistry, Faculty of Chemistry, University of Murcia, Campus of Espinardo, 30100 Murcia, Spain; 3Department of Chemical Engineering, Faculty of Chemistry, University of Murcia, Campus of Espinardo, 30100 Murcia, Spain

**Keywords:** autism, children, gastrointestinal symptoms, gut microbiota, systematic review

## Abstract

*Background and objectives*: Autism spectrum disorder (ASD) is a neurodevelopmental disorder characterized by impaired communication, social interaction disorder, and repetitive behavior. Dysbiotic gut microbiota (GM) could be a contributing factor to the appearance of ASD, as gastrointestinal (GI) symptoms are comorbidities frequently reported in ASD. As there is a lack of reviews about the role played by GM in the GI symptoms of ASD, this work aimed to carry out a systematic review of current studies comparing the GM of children with ASD and GI symptoms with those of healthy controls in the last six years. *Materials and Methods:* The systematic review was performed following the PRISMA guidelines. The databases chosen were Web of Science, Scopus, PubMed, and PsycINFO, and the keywords were (gut* OR intestine* OR bowel* OR gastrointestinal*) AND (microbiota* OR microflora* OR bacteria* OR microbiome* OR flora* OR bacterial* OR bacteria* OR microorganism* OR feces* OR stool*) AND (autistic* OR autism* OR ASD*). *Results*: A total of 16 articles were included. Ten articles performed correlations analysis between GI symptoms and ASD. Among those 10 articles, 7 found differences between the GI symptoms present in children with ASD and healthy controls. The most common GI symptom was constipation. Among the seven articles that found differences, three performed correlations analysis between GI symptoms and gut microbe abundance. *Candida*, *Prevotella*, *Streptococcus*, and *Veillonella* showed higher and lower abundance, respectively, in children with ASD and GI symptoms in more than one article. *Bacteroidetes*, *Firmicutes*, *Actinomyces*, *Dorea*, *Lactobacillus*, *Faecalibacterium prausnitzii*, and *Bacteroidetes*/*Firmicutes* ratios showed abundance discrepancies. *Conclusions*: It is still too early to draw a conclusion about the gut microbes involved in GI symptoms of ASD. Future research should consider the relationship between ASD behavior, GM, and GI symptoms in a multidisciplinary way and homogenize sample characteristics.

## 1. Introduction

Autism or autism spectrum disorder (ASD) is defined in the Diagnostic and Statistical Manual for Mental Disorders (Fifth Edition (DSM-5)) as a heterogeneous group of neurodevelopmental disorders featuring the three main characteristics of impaired communication, social interaction disorder, and repetitive behavior [1]. Autism includes Kanner’s autism [2], Asperger’s syndrome [3], and pervasive developmental disorder not otherwise specified (PDD-NOS), and is characterized by persistent deficits in social communication interaction and restricted–repetitive patterns of behavior, interests, or activities [1]. It is worth mentioning that people without neurological, medical, or psychiatric diagnoses are defined as neurotypical in the autism community [4].

Autism originally affected one in 10,000 in the 1950s, but now affects one in every 58 people in USA and around one in 30 in Korea, being more prevalent in men (4:1) [5], and possibly in recent immigrant populations (i.e., Somali expatriates in Sweden [6]), suggesting some link with industrialization. In this sense, there is a need for the scientific community to discover the root causes of ASD and the complex interactions between genetic, epigenetic, immunological, neurological, and environmental factors that might be contributing to the development and expression of ASD in sensitive populations [5,7].

People with ASD develop several comorbidities, including gut-related comorbidities such as gastrointestinal (GI) symptoms, increased permeability of the epithelial barrier in the gut, decreased expression of brush border disaccharidases in the gut epithelium, and altered gut microbiota (GM) composition; brain-related comorbidities such as altered expression of tight junction proteins in the blood–brain barrier and increased amounts of activated microglial cells; and other comorbidities such as mitochondrial dysfunction, fragile X syndrome, Rett syndrome, and tuberous sclerosis, and altered metabolite profiles in urine and blood [8].

Among the comorbidities developed by people with ASD, GI symptoms such as diarrhea, constipation, commutative diarrhea/constipation, abdominal pain, vomiting, reflux, or bloating are quite common, and they are correlated with the severity of the neurobehavioral disorder [9]. In this sense, people with ASD with GI symptoms exhibit more anxiety problems and other somatic complaints, together with less social interaction than ASD people without GI symptoms [8]. For example, Marler et al. [10] reported that constipation was found to be the most common GI symptom in ASD people, and that there is a connection between rigid–compulsive behavior and the occurrence of constipation. In addition, people with ASD and GI problems also show more tantrums, aggressive behavior, self-injury, and sleep disturbances, which might be an expression of abdominal discomfort [8].

The etiology of GI symptoms in ASD remains unknown, and everything seems to indicate that it is a combination of associated factors [9,11]. However, GI symptoms appear to be due, in part, to dysbiotic GM. In this sense, there is enough scientific evidence regarding the relationship between a dysbiotic GM and the development of diverse chronic diseases such as asthma, intestinal inflammation associated with the pathogenesis of obesity, type 2 diabetes mellitus [12], and other neurodevelopmental disorders [13].

The GM encompasses the set of bacteria that inhabit the (GI) tract. These bacteria have a dynamic symbiotic relationship that is both commensal and mutual throughout the host lifecycle [14]. At present, the microbiota–gut–brain axis is an explanatory model that attempts to relate the symptoms of ASD to the findings in neuroscience and bacteriology; it is defined as a bidirectional communication system between the neuronal, immune, endocrine, and metabolic pathways, but still requires a better understanding [15].

As discussed before, as GI symptoms can be related to dysbiotic GM, the use of pre/probiotics and microbiota transfer therapy (MTT) as therapeutic tools to treat people with ASD have been gaining greater interest in recent years, as they may help to restore normal GM, reduce inflammation, restore epithelial barrier function, and possibly improve some behavioral symptoms associated with ASD [16,17,18]. For example, Shaaban et al. [19] observed beneficial effects on both behavioral and GI manifestations in Egyptian ASD children with GI symptoms after administering *Lactobacillus acidophilus*, *Lactobacillus rhamnosus*, and *Bifidobacteria longum*, and Kang et al. [18] reported improvements in the microbiome and both GI and ASD symptoms in American ASD children with GI symptoms after MTT. These improvements were sustained at least 8 weeks after treatment. In addition, it has been reported that diet also has a strong correlation with GI symptoms in children with ASD [20], and GM appears to be affected by diet and has a modulatory role in several disease states. For example, besides pre/probiotics, dietary polyphenols, which are often indigestible, may also positively influence GM [21]. In one case, Sprague Dawley rats with hepatic steatosis who were administered a high-fat diet were later given curcumin, which not only restored intestinal barrier integrity (increasing the expression of tight junction proteins ZO-1 and occluding), but also markedly altered the overall composition of the GM towards that of lean rats maintained on a normal diet [22]. Of note, a strong correlation between GI symptoms and diet problems in ASD children, especially idiosyncratic feeding behavior, has been also reported, and ASD children suffering from multiple GI symptoms tend to be those who also have dietary problems [23].

Traditionally, characterization of GM has been carried out using culture-dependent techniques. One of the great limitations of these techniques is that it is impossible to ensure that the phenotypic behavior of a microorganism in the laboratory is identical to its behavior in vivo, due in part to the processes of metabolic symbiosis between microorganisms. In addition, the number of microorganisms that can be characterized with this technique is limited [24]. However, the emergence and rapid development of culture-independent or molecular techniques has made culture-dependent techniques nearly obsolete, and the characterization of GM is currently largely carried out using culture-independent techniques, as they allow scientists to easily identify a large proportion of bacterial diversity and provide rapid results [25,26]. This has led to many studies that compare the diversity of the GM of ASD children with healthy controls [27,28,29,30,31], of which a bibliographic review can be carried out. In this sense, several bibliographic reviews about GM and ASD can be found in the scientific literature. For example, Hughes et al. [32] and Liu et al. [33] studied the dysbiosis of GM in ASD, and Andreo-Martínez et al. [34] studied the gut microbes belonging to the Fungi kingdom in people with ASD. In addition, there have been few studies that have analyzed the correlation between GM composition and GI symptoms in people with ASD. For example, in a meta-analysis about GI symptoms in ASD [9], performed on studies published until 2012, there was only one study [35] which reported that GI symptoms in people with ASD were correlated with high levels of *Clostridium* (*p* = 0.001). However, to the author’s best knowledge, there have been no bibliographic reviews that have attempted to relate the gut microbes involved in the GI symptoms of people with ASD. Therefore, the aim of the present work was to carry out a systematic review on the studies that have compared the GM of people with ASD and GI symptoms with those of healthy controls, continuing the meta-analysis line of McElhanon, McCracken, Karpen, and Sharp [9], during the last six years, to try to find some bacterial abundance patterns and to identify future research strategies.

## 2. Materials and Methods

The design of the present systematic review was performed following the Preferred Reporting Items for Systematic Reviews and Meta-Analyses (PRISMA) guidelines [36]. This systematic approach under the PRISMA perspective covers the developments in the field of the GM of people with ASD and GI symptoms by providing critical analysis of key parameters, such as the role that gut microbes play in the GI symptoms of people with ASD, and gut microbe abundance of people with ASD with GI symptoms.

The bibliographic search was performed for works published between 2012 and February 2019, and the four comprehensive databases chosen to carry out the bibliographic search were: Web of Science, Scopus, PubMed, and PsycINFO. The Boolean strings entered to carry out the search were: (gut* OR intestine* OR bowel* OR gastrointestinal*) AND (microbiota* OR microflora* OR bacteria* OR microbiome* OR flora* OR bacterial* OR bacteria* OR microorganism* OR feces* OR stool*) AND (autistic* OR autism* OR ASD*), and the searches included works published in all language. Scopus search options were: title, abstract, and keywords; Web of Science search option was “theme,” while for PsycINFO and PubMed, the search option selected was all fields.

The inclusion criteria were: (1) articles with human populations; (2) articles published from 2012 to February 2019; and (3) articles comparing the GM of children with ASD and GI symptoms with those of control groups. The exclusion criteria were: (1) descriptive and systematic reviews; (2) books and book chapters; (3) editorial material, letters to the editor, shorts reports, and proceedings of conferences; (4) animal model and in vitro articles; (5) other diseases; (6) articles studying only metabolites in blood, plasma, or urine; and (7) articles published in a language other than English.

The 1540 works obtained by the four databases were passed through EndNote X7 software (Thomson Reuters, New York, USA) to delete possible duplicated works. According to the PRISMA methodology, A.E.M.-G. and P.A.-M. formed the review team and independently screened titles, abstracts, and full texts of the works for potential inclusion. These two authors evaluated them according to the inclusion or exclusion criteria, and disagreements on whether a given reference should be included or not were resolved through discussion. In addition, the reference lists of the selected works were also examined in search of potentially relevant papers, not finding any additional article in this step. It is noteworthy that some of the works could be included in more than one elimination group, but the final criteria were agreed by the review team. Finally, a total of 16 articles were found to be eligible for the present systematic review and they are shown in Table 1. The risk of bias for each included study was evaluated using the Methods Guide for Comparative Effectiveness Reviews [37]. The forms of bias considered are presented in Table 2.

Figure 1 shows the process of identifying articles for inclusion in the present systematic review.

## 3. Results

Table 1 shows the sixteen studies included in the present systematic review and the results of the analysis of the samples in children with ASD and GI symptoms.

### 3.1. Assessment of Risk of Bias for Each Included Article

The 16 articles were identified and assessed as medium to high quality (Table 2). When assessing the quality of selection, the studies presented some limitations in terms of sample size, information on comorbidity of neurodevelopmental disorders, diet of study participants, blinding of participants and personnel, and mean scores and standard deviations in the analysis of study data. Many of these studies were also open-label trials and relied on qualitative, self-reported questionnaires and surveys to gauge treatment response, inviting potential bias into the studies. There might also be various difficulties experienced by the parents in evaluating these aspects, especially owing to the communication deficits typical of children with ASD. More randomized, controlled studies with a larger study population and the use of clinician scoring may lead to more robust studies and results [17].

### 3.2. Relationship between GI Symptoms and ASD

Among the 16 articles found in the present systematic review, ten of them performed analysis to try to find correlations between GI symptoms and ASD. Seven articles found statistically significant differences between the occurrence of GI symptoms in children with ASD and healthy controls [28,30,31,42,43,44,48], and three articles found no such differences [27,39,41]. The remaining six articles did not perform statistical analysis to analyze correlations between GI symptoms and ASD [29,38,40,45,46,47].

The most common GI symptom in children with ASD was found to be constipation, results in line with those reported elsewhere [10]. In this sense, five articles reported statistically significant differences in children with ASD suffering constipation compared to healthy controls [30,42,43,44,48].

### 3.3. Relationship between Gut Microbe Abundance and GI Symptoms in Children with ASD

Three articles tried to find correlations between the GI symptoms of children with ASD and gut microbe abundance [31,42,43]. Two of those articles found correlations between certain gut microbes and GI symptoms, specifically constipation [42,43], and irritable bowel syndrome, functional constipation, aerophagia, and abdominal migraine [43]. In light of those results, some genera of bacteria are involved in constipation, but the results between the studies are not coincident (Table 1). The remain article reported no correlation between *Desulfovibrio* and GI symptoms [31].

### 3.4. Gut Microbiota Dysbiosis found in children with ASD and GI Symptoms

Seven articles reported statistically significant differences in the presence of GI symptoms in children with ASD compared to healthy controls [28,30,31,42,43,44,48]. These articles also showed significant differences in the abundance of some gut microbes in children with ASD suffering GI symptoms (Table 1) compared to their respective control groups. The gut microbes that were found by more than one article to show significant differences in abundance are listed below.

*Candida* was found to be the only gut microbe that showed significantly higher abundance in children with ASD and GI symptoms compared to their control groups by more than one article [30,42].

Lower abundance of *Prevotella* [27,44], *Veillonella* [42,48], and *Streptococcus* [44,48] in children with ASD and GI symptoms compared to their control groups was also found to be statistically significant.

In addition, some gut microbes showed discrepancies at a statistically significant level regarding their abundance in children with ASD and GI symptoms compared to their control groups. Specifically, at phylum level, *Bacteroidetes* [42,48] and *Firmicutes* [31,48]; at genus level, *Actinomyces* [44,48], *Dorea* [42,43], and *Lactobacillus* [30,31,42]; and at species level, *Faecalibacterium prausnitzii* [28,43]. The *Bacteroidetes*/*Firmicutes* ratio also showed discrepancies at a statistically significant level regarding their abundance in children with ASD and GI symptoms compared to their control groups [31,42,48].

### 3.5. Limitations

An important limitation of the studies is the absence of a psychometric analysis of the relationship between the severity of behavioral ASD symptoms with GM abundance and GI symptoms. Similarly, none of the studies indicated whether children with ASD presented a diagnosis of intellectual disability, although cognitive difficulties are a determining factor in the severity of ASD symptoms [49].

On the other hand, the present systematic review has the same limitations as other systematic reviews, such as the databases used, the established inclusion/exclusion criteria, database selection, keywords, or time-frame chosen. In addition, the lack of sufficient articles made it impossible to combine the results into a meta-analysis [9].

## 4. Discussion

The present systematic review found 16 articles that studied the GM in children with ASD and GI symptoms, and, as discussed before, among the 10 articles that analyzed the correlation between GI symptoms and ASD, 7 of them found significant differences. These results are similar to those reported by Adams et al. [50], who found a strong correlation between the 6-GSI score and the Autism Treatment Evaluation Checklist score (Pearson’s correlation coefficient *r* = 0.60; *p* < 0.001). Those results are also in line with a meta-analysis which included 15 articles comparing heterogeneous groups (siblings, ASD and healthy control) [9]. Among the 15 selected articles, 10, 12, 9, and 8 reported general GI concerns (*p* < 0.0001), diarrhea (*p* < 0.0001), constipation (*p* < 0.0001), and abdominal pain (*p* = 0.016), respectively. In conclusion, there seems to be evidence that there is a greater prevalence of GI symptoms among children with ASD compared with control children.

On the other hand, three studies [31,42,43] analyzed the correlation between gut microbe abundance and GI symptoms in children with ASD and GI symptoms. Although the studies are very promising, there is not yet enough scientific evidence to determine the relationships between gut microbe abundance and GI symptoms in children with ASD, as the gut microbes found to be correlated with GI symptoms in children with ASD were different between the different studies (Table 1). In this sense, the results are very premature, and the number of articles found was very low. Therefore, as discussed before, it was not possible to perform a meta-analysis of the relationships between gut microbes and GI symptoms in children with ASD [9].

Regarding the abundance of some gut microbes in children with ASD and GI symptoms, *Candida* was found to be more abundant by two studies [30,42], *Candida albicans* being the most frequently identified species [30]. In fact, it has been reported that a high abundance of *Candida albicans* can cause lower absorption of carbohydrates and minerals, and can generate higher toxins levels in the GI tract that can contribute to some ASD behaviors [51]. It has been also reported that the immunological responses to *Candida* in the GI tract can be modulated by some *Lactobacillus* species [42,52]. In this sense, it is possible that a dysbiotic GM in ASD may lead to a *Candida* population expansion, as can be observed by the lower abundance of *Lactobacillus* found in Italian children with ASD and GI symptoms compared to their control group [30]. However, a higher abundance of both *Lactobacillus* and *Candida* was also found in Italian children with ASD and GI symptoms compared to their control group [42]. Therefore, these results are not conclusive, and, as can be observed, there is a clear lack of investigation of the whole Fungi kingdom in children with ASD, and little is still known about the role of *Candida* and other types of fungi in both GI and ASD symptoms [34].

Lower abundances of three gut microbe genera (*Prevotella* [27,44], *Veillonella* [42,48], and *Streptococcus* [44,48]) in children with ASD and GI symptoms were also reported by more than one article included in the present systematic review. Lower abundance of *Prevotella* may act as an indicator of Westernization, and this fact can also result in an altered immune system [53]. In addition, the presence of *Prevotella* together with other *Bacteroidetes* is also associated with colon health, and it has been hypothesized that ASD can be triggered by a dysbiosis where *Prevotella* decreases and *Sutterella* increases [54]. In this sense, the lower abundance of *Prevotella* found in American children with ASD was associated with the presence of ASD symptoms instead of GI symptoms [27]. These authors did not find this association in a later work [28], although they found that GI symptoms were significantly more severe in children with ASD compared to their control group.

*Streptococcus*, together with *Lactobacillus*, *Bifidobacterium*, and *Lactococcus*, are lactate producing bacteria [48]. It is known that children with ASD show elevated levels of both lactate and GI symptoms [55], suggesting an elevation of glycolysis through the phenomenon of aerobic glycolysis in ASD, since the dysregulation of this balance has been also been proposed as a trigger for ASD [56]. In this sense, as *Veillonella* is also able to ferment lactate, a lower abundance of *Veillonella* may also disturb the fermentation of lactate in children with ASD [48]. However, it cannot be stated that the elevated levels of lactate in people with ASD and GI symptoms can be attributed only to the GM, since, as discussed before, there are discrepancies in the abundance of other lactate-producing bacteria, such as *Lactobacillus*. Regarding the implications of *Veillonella* in GI symptoms presented by children with ASD, *Veillonellaceae*, together with *Prevotellaceae*, *Prevotella*, and *Coprococcus*, has been included in a probiotic mixture for treating ASD [57], and it is known that the differences in GM composition (especially in *Veillonella*) of children with ASD may be a consequence of diet [27,46].

The discrepancies found, at a statistically significant level, in the abundance of *Bacteroidetes* [42,48] and *Firmicutes* [31,48] phyla; *Actinomyces* [44,48], *Dorea* [42,43], and *Lactobacillus* [30,31,42] genera; *Faecalibacterium prausnitzii* species 28,43]; and *Bacteroidetes*/*Firmicutes* ratio [31,42,48] can be explained by several factors such as ASD heterology, age of participants, nature of control groups, sample location, small sample sizes, different nationalities, inter-individual differences, or different bacterial identification methods [14,31,53,58]. For example, one study used a culture-dependent method to detect *Lactobacillus* [30], and it was found to be less abundant in children with ASD and GI symptoms compared to the control group. However, two studies [31,42] that used culture-independent methods found opposite results, and although some authors have postulated that both methods yield similar results [59], culture-independent methods are relatively new and they are still developing [26]. Another study found a higher abundance of *Dorea* in stool samples of children with ASD and GI symptoms [42], while other studies found opposite results taken from rectum mucosal biopsy samples, and it is known that the place where the sample is taken can be relevant for mammalian physiology [14], as there are differences in microbial composition between stool and the GM [60,61]. This was confirmed in a larger-scale study showing that the populations recovered in the stool seemed to combine bacteria derived from the mucous membrane and the luminal part of the intestine, either adherent or not adherent to transient organic matter [62]. It is assumed that microorganisms seen in the stool reflect the microbiology of the colon, particularly the descending colon and rectum. Finally, it is important to highlight that there are still existing gaps in knowledge regarding the interactions between the microbiome and the host in vivo—and the pathways of metabolites—and how their metabolites influence the microenvironment. Therefore, further mechanistic studies involving "omics" technologies, as adapted from previous studies [63], might help shed light on these questions, and future studies should try to homogenize, as much as possible, the characteristics of the samples to be compared, including the place where the sample is taken from and bacterial detection methods, among others.

## 5. Conclusions

Although recent scientific literature has provided evidence of the relationship between GI symptoms and ASD, it is still too early to draw a conclusion about the gut microbes involved in GI symptoms of children with ASD, due to the limited number of studies reporting correlations between them. Furthermore, the correlation between certain gut microbes and GI problems do not determine the causality of the symptoms. However, it can be said that children with ASD and GI symptoms show higher abundance of *Candida*, lower abundance of *Prevotella*, *Veillonella*, and *Streptococcus*, and also show discrepancies in the abundance of other gut microbes due to ASD heterology, age of participants, nature of control groups, sample location, small sample sizes, different nationalities, inter-individual differences, or different bacterial identification methods. In addition, we consider that ASD has a wide phenotypic variability, so future studies should consider the relationship between GM, GI symptoms, and behavior of ASD in a more integral and multidisciplinary way. Finally, we argue that future research should homogenize sample characterization in order to develop a meta-analysis on the GM involved in ASD children with GI symptoms.

## Figures and Tables

**Figure 1 medicina-55-00408-f001:**
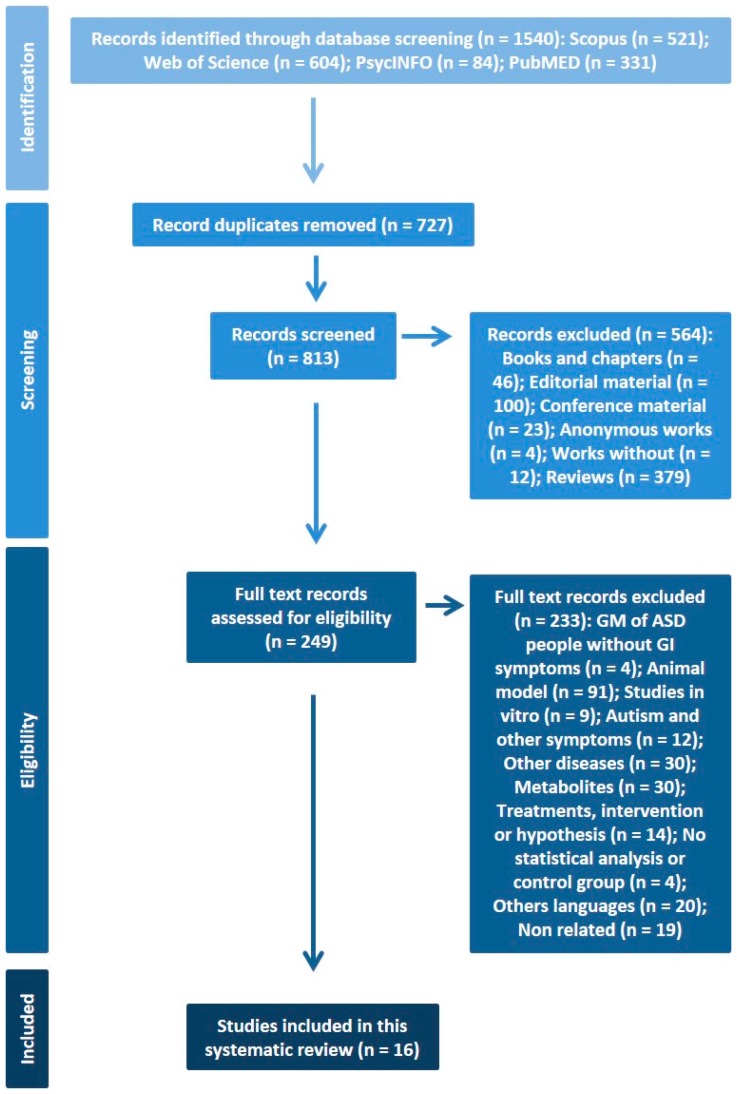
Flowchart showing the process of identifying relevant studies for the present systematic review.

**Table 1 medicina-55-00408-t001:** Characteristics of the selected studies on gut microbiota and children with ASD and GI symptoms.

Reference	Subjects		Method	Findings		Limitations
	Experimental Group	Control Group		Bacteria	Correlation	
[38]	ASD (n = 23) divided into ASD w GI symptoms (12/23) and ASD w/o GI symptoms (11/23) Range Age 3–10 y (15/23 aged 3–5 y; 6/23 aged 6–7 y; 2/23 aged 8–10 y) All males	HC w GI symptoms (n = 9) Range Age 3–10 y (7/9 aged 3–5 y; 1/9 aged 6–7 y; 1/9 aged 8–10 y). All males	Pyrosequencing of the 16S rRNA gene pan-bacterial (V2 region) on ileal and ceca mucosal biopsy samples PCR-based detection of *Sutterella* 16S rRNA gene sequences (V6–V8 region and C4–V8 region) on ileal and ceca mucosal biopsy samples	*Sutterella* ↑ in ASD w GI symptoms vs. HC w GI symptoms (ilea *p* = 0.022/ceca *p* = 0.037)	No correlations between GM and GI symptoms were studies	ASD severity was not indicated Was not indicated whether the ASD subjects had intellectual disability
[39]	ASD (n = 51) divided into ASD w GI symptoms (28/51) and ASD w/o GI symptoms (23/51) Age 2–12 y 42 males 9 females	NT Sib (n = 53) Age 2–12 y 19 males 34 females	Pyrosequencing (bTEFAP) of the 16S rRNA gene bacterial (V1–V3 regions) on fecal samples	No difference in microbiome between total ASD vs. NT Sib. No difference regarding severity ASD in the bacterial composition. No difference regarding GI symptoms dysfunction in the bacterial composition (No differences within the ASD group compared to both with and without GI symptoms groups)	No metabolites were studied	Was not indicated whether the ASD subjects had intellectual disability
[27]	ASD w GI symptoms (n = 20) Age 3–16 y 18 males 2 females	NT w GI symptoms (n = 20) Age 3–16 y 17 males 3 females	Pyrosequencing (bTEFAP) of the 16S rRNA gene bacterial (V2–V3 regions) on fecal samples Quantitative real-time PCR for *Prevotella*	*Prevotella* (*p* = 0.04), *Coprococcus* (*p* ≤ 0.06) and unclassified *Veillonellaceae* (*p* = 0.04) ↓ in ASD vs. NT. Microbial changes were more closely linked to the presence of autistic symptoms rather than to the severity of GI symptoms and specific diet/supplement regimens	No correlations between GM and GI symptoms were studies	Was not indicated whether the ASD subjects had intellectual disability
[40]	ASD (n = 23) w or w/o GI symptoms Age 3–18 y Unrevealed sex	HC (n = 9) and NT sib (n = 22) w or w/o GI symptoms Age 3–18 y Unrevealed sex	qPCR for *Sutterella*, *Ruminococcus torques* and *R. gnavus* on fecal samples.	*Sutterella* ↑ (*p* = 0.044) in ASD vs. NT Sib. *Sutterella* ↑ (*p* = 0.05) in ASD vs. HC. *Sutterella* ↑ (*p* = 0.047) in NT Sib vs. HC. *Ruminococcus gnavus* ↑ (*p* = 0.046) in NT Sib vs. HC. *Ruminococcus torques* ↑ (*p* = 0.008) in ASD w GI symptoms (n = 9) vs. ASD w/o GI symptoms (n = 14).	No correlations between GM and GI symptoms were studies	ASD severity was not indicated Was not indicated whether the ASD subjects had intellectual disability
[41]	ASD (n = 59) divided into ASD w GI symptoms (25/59) and ASD w/o GI symptoms (34/59) Age 7–14 y 52 males 7 females	NT Sib (n = 44) divided into NT Sib w GI symptoms (13/44) and NT Sib w/o GI symptoms (31/44) Age 7–14 y 21 males 23 females	qPCR for total bacteria, *Sutterella* subgroup, *Bacteroidetes* subgroup, *Prevotella*, *C. coccoides-E. rectales* subgroup, *Faecalibacterium prausnitzii* and *Escherichia coli* subgroup on fecal samples Sequencing of the 16S rRNA gene bacterial (V1–V2 and V1–V3 regions) on fecal samples	No difference between total ASD vs NT Sib in bacterial frequency. Increased prevalence of functional constipation GI symptoms in ASD children compared to NT siblings	Increased prevalence of functional constipation GI symptoms in ASD children compared to NT siblings.	ASD severity was not indicated Was not indicated whether the ASD subjects had intellectual disability
[31]	ASD (n = 10) divided into ASD w GI symptoms (9/10) and ASD w/o GI symptoms (1/10) Age 2–9, 9 males and 1 female Sib (n = 9) divided into Sib w GI symptoms (7/9) and Sib w/o GI symptoms (2/9) Age 5–17, 7 males and 3 females	HC (n = 10) divided into HC w GI symptoms (6/10) and HC w/o GI symptoms (4/10) Age 2–11, 10 males	qPCR for *Bacteroidetes*, *Firmicutes*, *Bifidobacterium*, *Lactobacillus*, *Clostridium* cluster 1, *S. thermophiles*, *Desulfovibrio* on fecal samples	*Bacteroidetes*/*Firmicutes* ratio↓ in ASD vs. HC (*p* < 0.05); ↓ Sib vs. ASD (*p* < 0.05); ↓ Sib vs. HC (*p* < 0.05) *Firmicutes* ↑ in Sib vs. HC *Lactobacillus* ↑ in ASD vs. HC (*p* < 0.05) *Bifidobacterium* ↑ in ASD vs. Sib (*p* < 0.05)	*Desulfovibrio* with ASD severity in the ADI restricted/repetitive behavior subscale score. There is a correlation of the autism severity with the severity of GI dysfunction. (*p* = 0.01) No correlation of GI symptoms and *Desulfovibrio* abundance.	Was not indicated whether the ASD subjects had intellectual disability
[42]	ASD (n = 40) divided into severe ASD (36/40) and moderately severe ASD (4/40) 5 constipated 29 non-constipated Average age 11.1 ± 6.8 31 males 9 females	NT (n = 40) 11 constipated 29 non-constipated Average age 9.2 ± 7.9 28 males 12 females	Pyrosequencing of the 16S rRNA gene bacterial (V3–V5 regions) and the internal transcribed spacer (ITS) for fungal (ITS1 rDNA region) on fecal samples	*Firmicutes*/*Bacteroidetes* ratio↑, *Bacteroidetes* ↓, *Veillonella* ↓, *Alistipes* ↓, *Bilophila*↓, *Dialister*↓, *Parabacteroides*↓ in ASD vs. NT (*p* < 0.005) *Lactobacillus* ↑, *Dorea* ↑, *Corynebacterium* ↑, *Collinsella* ↑, *Candida* ↑ in ASD vs. NT (*p* < 0.001)	*Escherichia*/*Shigella* and cluster XVIII with GI symptoms or constipation (*p* < 0.05)	Was not indicated whether the ASD subjects had intellectual disability
[30]	Severe ASD (n = 47) w/o or w GI symptoms Average age 6 ± 2.8 y 40 males 7 females	HC (n = 33) w/o or w GI symptoms Average age 7.3 ± 3.1 y 24 males 9 females	Examination and culture of fecal samples: (a) morphological examination, (b) microscopic examination staining, (c) search for toxins a/b of *Clostridium difficilis*, (d) bacterial/yeast culture and (e) identification of bacteria and yeast colonies by VITEK 2 microbial identification system	*Candida* aggressive form (pseudo-hyphae presenting) *Candida* ↑ in ASD vs. HC (*p* = 8.67 × 10^−6^) *Lactobacillus* ↓ in ASD vs. HC (*p* = 7.28x10^−4^) *Clostridium* ↓ in ASD vs. HC (*p* = 0.01)	GI symptoms in 70.2% ASDs and no controls, with a mild correlation by multivariate analyses of constipation and alternating bowel versus increased permeability to lactulose.	Was not indicated whether the ASD subjects had intellectual disability
[43]	ASD w GI symptoms (n = 14) Age 4–13 y 14 males	NT w GI symptoms (n = 15) Age 3–18 y 12 males 3 females NT w/o GI symptoms (n = 6) Age 3–18 y 6 males	Sequencing of the 16S rRNA gene (V1–V3 and V4 regions) on rectum mucosal biopsy samples	*Blautia* ↓ (*p* = 0.02), *Dorea* ↓ (*p* = 0.006), *Sutterella* ↓ (*p* = 0.025) in ASD w GI symptoms vs. NT w GI symptoms *Clostridium lituseburense* ↑ (*p* = 0.002), *Lachnoclostridium bolteae* ↑ (*p* = 0.017), *Lachnoclostridium hathewayi* ↑ (*p* = 0.03), *Clostridium aldenense* ↑ (*p* = 0.03), and *Flavonifractor plautii* ↑ (*p* = 0.03), in ASD w GI symptoms vs. NT w GI symptoms and NT w/o GI symptoms *Faecalibacterium prausnitzii* ↑, *Roseburia intestinalis* ↑, *Oscillospira valericigenes* ↑, and *Bilophila wadsworthia* ↑ (*p* < 0.05) in NT w GI symptoms vs. NT w/o GI symptoms	IBS with ↑ *Clostridium aldenense* (*p* = 0.04); Functional constipation with ↓ *Flavonifractor plautii* (*p* = 0.03), *Bacteroides eggerthii* (*p* = 0.02), *Bacteroides uniformis* (*p* = 0.04), *Faecalibacterium prausnitzii* (*p* = 0.013), *Clostridium clariflavum* (*p* = 0.03); Aerophagia with ↑ *Clostridium aldenense* (*p* = 0.03), ↓ in *Blautia luti* (*p* = 0.003), *Bifidobacterium adolescentis* (*p* = 0.01), *Eubacterium ventriosum* (*p* = 0.05), *Anoxystipes fissicatena* (*p* = 0.02), *Coprococcus comes* (*p* = 0.04), *Eubacterium ramulus* (*p* = 0.006), and *Phascolarctobacterium faecium* (*p* = 0.04); Abdominal migraine with ↓ in *Akkermansia muciniphila* (*p* = 0.03), *Coprococcus catus* (*p* = 0.007), *Odoribacter splanchnicus* (*p* = 0.05), *Clostridium lactatifermentans* (*p* = 0.03) and *Ruminococcus lactaris* (*p* = 0.03) Serotonin with *Lachnoclostridium bolteae* (*p* = 0.002), *Lachnoclostridium hathewayi* (*p* = 0.003) and *Flavonifractor plautii* (*p* = 0.001)	ASD severity was not indicated Was not indicated whether the ASD subjects had intellectual disability
[29]	ASD w GI symptoms (n = 33) Age 2–9 y Unidentified sex	HC w/o GI symptoms (n = 13) Age 2–9 y Unidentified sex	Selective culture methods for *Clostridium* and for *Clostridium perfringens* strains on fecal samples: Brucella and CDC agar. PCR for *Clostridium perfringens* toxin genes: alpha (cpa), beta (cpb), beta 2 (cpb2), epsilon (ctx), iota (iA), and enterotoxin (cpe)	*Clostridium perfringens* ↑ in ASD w GI symptoms vs. HC (*p* = 0.03)	No correlations between GM and GI symptoms were studies	ASD severity was not indicated Was not indicated whether the ASD subjects had intellectual disability
[44]	ASD (n = 21) w GI symptoms Age 14.4 ± 1.1 y 19 males 2 females	HC (n = 19) w GI symptoms Age 16 ± 1.2 y 10 males 9 females	Pyrosequencing of the 16S rRNA gene bacterial (V1–V3 regions) on duodenal biopsies samples from the second part of the duodenum	No differences in microbiome diversity (alpha and beta) *Burkholderia* ↑ (*p* = 0.03) and *Neisseria* ↓ (*p* = 0.01) in ASD vs. HC *Bacteroides vulgatus* ↓ (*p* = 0.005), unidentified *Bacteroides* ↓ (*p* = 0.04), and *Escherichia coli* ↓ (*p* = 0.05) in ASD vs. HC *Oscillospira*, *Actinomyces*, *Peptostreptococcus*, and *Ralstonia* ↑ (*p* < 0.05) in ASD vs. HC *Devosia*, *Prevotella*, *Bacteroides*, and *Streptococcus* ↓ (*p* < 0.05) in ASD vs. HC	↑ frequency of constipation in ASD vs. HC (*p* < 0.005)	ASD severity was not indicated Was not indicated whether the ASD subjects had intellectual disability
[45]	ASD (n = 29) w GI symptoms (49 isolated strains of *Clostridium perfringens*) Age 3.5–18 y 23 males 9 females	HC (n = 17) (30 isolated strains of *Clostridium perfringens*) Obese children (n = 24) (32 isolated strains of *Clostridium perfringens*) Unrevealed age and sex	Selective culture method for fecal samples: Columbia blood and reinforced clostridial agar under anaerobic conditions. Hemolysis test, lecithinase, and lipase production on egg yolk agar, and identified with use of ANC cards in VITEK 2 compact. Subcultured in BHI broth and Gene MATRIX DNA Purifi cation Kit by DNA Gdansk for isolation of DN Multiplex PCR for toxin alpha (*cpa*), toxin beta (*cbp*), enterotoxin (*ecpe*), iota-toxin (*cpiA*), epsilon toxin (*etx*) genes	The *cpa* gene encoding alpha toxin was present in all 111 (100%) strains The *cpb2* gene encoding beta2 toxin was found in: 45/49 (91.8%) strains isolated from ASD children, 17/30 (56.7%) strains isolated from healthy subjects (*p* < 0.001), and 12/32 (37.5%) strains isolated from obese children (*p* < 0.001) *Clostridium perfringens* (*cpb2* gene) was detected in: 27/29 (93.1%) ASD, 10/17 (58.8%) HC (*p* < 0.008), and 11/24 (45.8%) obese children (*p* < 0.001) No differences between HC and obese children (*cpb2* and *Clostridium perfringens* with *cpb2*)	No correlations between GM and GI symptoms were studies	ASD severity was not indicated Was not indicated whether the ASD subjects had intellectual disability A small number of studied patients and strains
[28]	ASD (n = 23) divided into ASD w GI symptoms (21/23) and ASD w/o GI symptoms (2/23) Age 4–17 y 15 males 6 females	NT (n = 21) divided into NT w GI symptoms (10/21) and NT w/o GI symptoms (11/21) Age 4–17 y 22 males 1 female	Pyrosequencing of the 16S rRNA gene bacterial (V2–V3 regions) on fecal samples	Gut microbial diversity (alpha) ↓ (*p* < 0.001) and relative abundances of phylotypes most closely related to *Prevotella copri* ↓ (*p* < 0.04) in ASD *Faecalibacterium* ↓ (*p* < 0.01) and *Haemophilus* ↓ (*p* < 0.05) in ASD vs. NT *Feacalibacterium prausnitzii* ↓ (*p* < 0.01) and *Haemophilus parainfluenzae* ↓ (*p* < 0.05) in ASD vs. NT	GI symptoms were significantly more severe for children with ASD compared to controls (ATEC subscale = *p* < 0.01)	ASD severity was not indicated Was not indicated whether the ASD subjects had intellectual disability
[46]	ASD (n = 30) divided into severe ASD (28/30) and moderate ASD (2/30) w GI symptoms Age 3–16 y 28 males 2 females BMI 6.9–20.5	HC mostly Sib or blood relatives to the ASD children (n = 24) w/o GI symptoms Age 3.5–16 y 15 males 9 females BMI 13.4–31	Sequencing of the 16S rRNA gene bacterial (V3 region) on fecal samples	*Firmicutes* ↑ in ASD vs. HC Sib (*p* < 0.05) *Prevotellaceae* ↓ and *Veillonelleaceae* ↑ in ASD vs. HC Sib *Lactobacillaceae* ↑ (*p* = 0.018), *Bifidobacteriaceae* ↑ (*p* = 0.0054), and *Veillonellaceae* ↑ (*p* = 0.008) in ASD vs. HC Sib *Erysipelotrichaceae* ↑ (*p* = 0.0005), *Enterococcaceae* ↑ (*p* = 0.0127), and *Desulfovibrionaceae* ↑ (p = 0.03) in ASD vs. HC Sib *Bifidobacterium* ↑ (*p* = 0.005), *Lactobacillus* ↑ (*p* = 0.018), *Megasphaera* ↑ (*p* = 0.0008), and *Mirsuokella* ↑ (*p* = 0.007) in ASD vs. HC Sib 99% of *Lactobacillus* was *Lactobacillus ruminis* in ASD group	No correlations between GM and GI symptoms were studies	Was not indicated whether the ASD subjects had intellectual disability
[47]	ASD w GI symptoms (blood: n = 20, 15 males, 5 females) (stool: n = 21, 17 males, 4 females) ASD w/o GI symptoms (blood: n = 26, 19 males, 7 females) (stool: n = 29, 25 males, 4 females) Age 3–12 y	HC w GI symptoms (blood: n = 6, 5 males, 1 female) (stool: n = 7, 6 males, 1 female) HC w/o GI symptoms (blood: n = 35, 24 males, 11 females) (stool: n = 34, 32 males, 2 females) Age 3–12 y	Sequencing of the 16S rRNA gene bacterial (V3–V4 regions) on fecal samples	*Bacteroidaceae* ↑, *Lachnospiraceae* ↑, *Ruminococcaceae* ↑, and *Prevotellaceae* ↑ in ASD w GI symptoms vs. HC w GI symptoms Il-5, IL-15, and IL-17 ↑ in ASD w GI symptoms vs. ASD w/o GI symptoms (after exposure to the TLR-4 agonist LPS) TGFbeta1↓ in ASD w GI symptoms vs. ASD w/o GI symptoms and HC w/o GI symptoms (*p* < 0.05) (under the majority of conditions examined)	Differences in the microbiome composition of children with ASD vs HC groups, irrespective of GI symptoms	ASD severity was not indicated. Was not indicated whether the ASD subjects had intellectual disability Limited sample size and younger age of the HC with GI symptoms group
[48]	ASD w GI symptoms (n = 35, 29 males, 6 females) Age 3–8 y	HC (n = 6, 5 males, 1 females) Age 3–8 y	Sequencing of the 16S rRNA gene bacterial (V3–V4 regions) on fecal samples.	*Firmicutes/Bacteroidetes* ratio↑ in ASD w GI symptoms vs. HC (*p* < 0.05) *Bacteroidetes* ↑ in ASD w GI symptoms vs. HC (*p* < 0.05) *Firmicutes* ↓ in ASD w GI symptoms vs. HC (*p* < 0.05) *Veillonella*, *Streptococcus*, *Escherichia*, *Actinomyces*, *Parvimonas*, *Bulleida*, and *Peptoniphilus* ↓ in ASD w GI symptoms vs. HC (*p* < 0.05)	Positive microbe-based link between periodontitis and ASD Negative microbe-based link between type 1 diabetes, constipation (*p* < 0.05), IBS, psoriasis, and ASD	ASD severity was not indicated. Was not indicated whether the ASD subjects had intellectual disability A small number of studied patients and strains

Note: w = with; w/o = without; ASD = Autism Spectrum Disorder; HC = Healthy control; GI = Gastrointestinal; NT = Neurotypical; NT Sib = Neurotypical siblings; Sib = siblings; y = years; ADI = Autism Diagnostic Interview; DHEA-S = dehydroepiandrosteronesulfate; IBS = Irritable Bowel Syndrome; BMI = Body Mass Index, FAA = Free Amino Acids; NT = HC; ATEC = Autism Treatment Evaluation Checklist.

**Table 2 medicina-55-00408-t002:** Review of author judgments on quality assessment for each included study.

Item	[38]	[39]	[27]	[40]	[41]	[31]	[42]	[30]	[43]	[29]	[44]	[45]	[28]	[46]	[47]	[48]
Clear stated aim	2	2	2	2	2	2	2	2	2	2	2	2	2	2	2	2
Appropriate study size	1	2	2	1	2	1	2	2	1	1	2	2	2	2	1	1
Identified and assessed	2	2	2	2	2	2	2	2	2	2	2	2	2	2	2	2
Comparability	2	2	2	2	2	2	2	2	2	2	2	2	2	2	2	2
Blinding of participants and personnel	0	0	0	0	0	0	0	0	0	0	0	0	0	0	0	0
Other bias (dietary intake controlled, reports on comorbidity and severity of ASD)	0	0	0	0	0	0	0	0	0	0	0	0	0	0	0	0
Adequate statistical analyses	2	2	2	2	2	2	2	2	2	2	2	2	2	2	2	2
TOTAL	9	10	10	9	9	9	10	10	9	9	10	10	10	10	9	9
Risk of bias	5	4	4	5	5	5	4	4	5	5	4	4	4	4	5	5
Overall risk of bias	M	M	M	M	M	M	M	M	M	M	M	M	M	M	M	M

0 = Not reported, 1 = not adequately assessed, 2 = adequately assessed; M = Medium (8–10); L = Low (9–14); H = High (7–1).

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
