# Peer review of "The Role of Gut Microbiota in Gastrointestinal Symptoms of Children with ASD"

_medicina, 2019, doi:10.3390/medicina55080408_

Round 1
Reviewer 1 Report
Authors performed a systematic review on gut microbiota and ASD.
- The works obtained from the four databases they used were 1540, however, only final 14 articles were suitable for inclusion for the systematic review. These articles (in addition, several articles did not study correlations between GM and GI symptoms) are too few to conclude a final outcome. I suggest, at least, to increase the range of the years of publication.
- According to authors, useful articles were 8, and 7 articles found positive correlations. It indicates, even if requiring more research, a significative trend forward these conclusions. Authors could briefly discuss it.
Reviewer 2 Report
Although the manuscript focuses on an interesting area for discussion, I found several areas of the paper considerably underdeveloped and the writing requires a close edit (ideally by a native English speaker) for language and grammar.
Specific comments:
Please do a close reading of the manuscript for language and grammar. Change "the route causes of ASD" (wrong word) to "the root causes of ASD".
Please change "This fact has led to the appearance of several studies" to "This has led to many studies".
Besides pre/probiotics, dietary polyphenols, which are often indigestible, may also positively influence gut microbiota (citation: ncbi.nlm.nih.gov/pubmed/30248988). In Sprague Dawley rats with hepatic steatosis (induced by a high-fat diet), curcumin not only restored intestinal barrier integrity (increased expression of tight junction proteins ZO-1 and occluding), it markedly altered the overall composition of the gut microbiota, towards that of lean rats maintained on a normal diet (citation: ncbi.nlm.nih.gov/pubmed/28341485). Gut microbiota appears to be affected by diet and has a modulatory role in several disease states.
Please provide the full electronic search strategy used to identify studies, including all search terms and limits for at least one database.
Why was the literature search limited to English language only? Can the authors give a reason why no grey literature was included in the analysis? This may be particularly important and relevant given the small number of studies found.
Were the inclusion criteria framed a priori – i.e. prior to the literature search?
No assessment for risk of bias.
Scientific names such as "Veillonellaceae" and "Bacteroidetes/Firmicutes" should be italicized.
An important point the authors did not address is that there are, in fact, important differences in microbial composition between stools and the gut microbiota (see: ncbi.nlm.nih.gov/pubmed/11571208 and ncbi.nlm.nih.gov/pubmed/16730943). This was confirmed in a larger-scale study showing that the populations recovered in the stools seem to combine bacteria derived from the mucous membrane and the luminal part of the intestine, either adherent or not adherent to transient organic matter (see: ncbi.nlm.nih.gov/pubmed/15831718). It is assumed that microorganisms seen in the stool reflect the microbiology of the colon, particularly the descending colon and rectum.
Please change "it is still early to draw a conclusion" to "it is still too early to draw a conclusion".
Please change "due to the limit number of studies" to "due to the limited number of studies".
Besides the fact that there are limited studies "reporting correlations between them", correlation also does not imply causation. This is important.
It is also important to highlight that there are still existing gaps in knowledge regarding the interaction between the microbiome and the host in vivo - and the pathway of its metabolites - and how their metabolites influence the microenvironment. Further mechanistic studies involving "omics" technologies, as adapted from previous studies (citation: ncbi.nlm.nih.gov/pubmed/30056340), might help shed light on these questions.
Round 2
Reviewer 2 Report
- Figure 1 requires some adjustments; please do not justify the text.
- As GI in itself is already an abbreviation, suggest authors avoid the use of the confusing abbreviation "GIS". It should be GI symptoms.
Author Response
Figure 1 requires some adjustments; please do not justify the text.
ANSWER:
Thanks for this comment. Following the recommendation of the reviewer, the text of figure 1 has been centered.
As GI in itself is already an abbreviation, suggest authors avoid the use of the confusing abbreviation "GIS". It should be GI symptoms.
ANSWER:
Thanks for this comment. Following the recommendation of the reviewer “GIS” has been replaced by “GI symptoms” in Table 1.